# Genetic inactivation of the β1 adrenergic receptor prevents cerebral cavernous malformations in zebrafish

**Wenqing Li\*, Sara McCurdy, Miguel A Lopez-Ramirez, Ho-Sup Lee, Mark H Ginsberg**

Department of Medicine, University of California, San Diego, San Diego, United States

## eLife Assessment

In this **important** study, the authors test the model that a type of vascular lesion caused by the inactivation of one gene in the cells that line blood vessels requires the activity of a second gene for the lesions to form. The evidence supporting the conclusions is **solid**.

**\*For correspondence:**
liwenqing753@gmail.com

**Competing interest:** The authors declare that no competing interests exist.

**Abstract** Previously, we showed that propranolol reduces experimental murine cerebral cavernous malformations (CCMs) and prevents embryonic caudal venous plexus (CVP) lesions in zebrafish that follow mosaic inactivation of *ccm2* (Li et al., 2021). Because morpholino silencing of the β1 adrenergic receptor (*adrb1*) prevents the embryonic CVP lesion, we proposed that *adrb1* plays a role in CCM pathogenesis. Here, we report that *adrb1*$^{-/-}$ zebrafish exhibited 86% fewer CVP lesions and 87% reduction of CCM lesion volume relative to wild type brood mates at 2dpf and 8–10 weeks stage, respectively. Treatment with metoprolol, a β1 selective antagonist, yielded a similar reduction in CCM lesion volume. *Adrb1*$^{-/-}$ zebrafish embryos exhibited reduced heart rate and contractility and reduced CVP blood flow. Similarly, slowing the heart and eliminating the blood flow in CVP by administration of 2,3-BDM suppressed the CVP lesion. In sum, our findings provide genetic and pharmacological evidence that the therapeutic effect of propranolol on CCM is achieved through β1 receptor antagonism.

## Introduction

Cerebral cavernous malformations (CCMs), accounting for 5–15% of cerebrovascular abnormalities, are characterized by blood-filled endothelial-lined cavities, and can cause seizures, headaches, neurological deficits, and recurrent stroke risk (*Leblanc et al., 2009*). Familial CCMs are due to heterozygous loss of function mutations in the *KRIT1*, *CCM2*, or *PDCD10* genes with a second mutation inactivating the normal allele in random brain endothelial cells (*Labauge et al., 2007*). We previously developed a zebrafish CCM model using CRISPR-Cas9 to inactivate *ccm2* in a manner that replicates the mosaic genetic background of the human disease (*Li et al., 2021b*). This zebrafish model exhibits two phenotypic phases: lethal embryonic caudal venous plexus (CVP) cavernomas at 2 days post-fertilization (dpf) in ~30% of embryos and histologically-typical CNS CCMs in ~100% of surviving 8-week-old fish (*Li et al., 2021b*). Both phases of this model, like their murine counterpart, depend on Krüppel-like factor 2 (KLF2), confirming shared transcriptional pathways. Furthermore, both phases of the zebrafish model exhibit similar pharmacological sensitivities (*Supplementary file 1*) to murine and human lesions. Thus, this zebrafish model offers a powerful tool for genetic and pharmacological analysis of the mechanisms of CCM formation.

Anecdotal case reports (*Moschovi et al., 2010*; *Reinhard et al., 2016*; *Goldberg et al., 2019*) and a recent phase 2 clinical trial have suggested that propranolol a non-selective β adrenergic receptor antagonist benefit patients with symptomatic CCMs (*Lanfranconi et al., 2023*). In previous studies, we observed this effect of propranolol in both zebrafish and murine models of CCM (*Li et al., 2021a*). Importantly, propranolol is a racemic mixture of R and S enantiomers and elegant work from the Bischoff lab has implicated the R enantiomer, which lacks β adrenergic antagonism, as the component that inhibits SOX18 thereby suppressing infantile hemangiomas (*Overman et al., 2019*). We previously found that the anti-adrenergic S enantiomer rather than the R enantiomer inhibited development of embryonic CVP cavernomas in *ccm2* CRISPR zebrafish (*Li et al., 2021a*). Furthermore, morpholino silencing of the gene that encodes the β1 (*adrb1*) but not β2 (*adrb2*) receptor also prevents CVP cavernomas (*Li et al., 2021a*), suggesting that the β1 adrenergic receptor (β1AR), which primarily impacts hemodynamics (*van den Meiracker et al., 1989*), contributes to the pathogenesis of CCM. Here we have inactivated the *adrb1* gene to eliminate the β1AR and observed the expected reduction of heart rate and contractility which resulted in reduced blood flow through the CVP. These *adrb1*$^{-/-}$ zebrafish were protected from embryonic CVP cavernomas and, importantly, also exhibited much reduced number and volume of adult brain CCM. Furthermore, a β1-selective antagonist, metoprolol, also inhibited adult CCM suggesting that β1AR-specific antagonists may be useful for CCM treatment and will carry less potential for β2 AR-related side effects such as bronchospasm (*Maclagan and Ney, 1979*).

## Results

### β1AR is important in for development of CVP cavernomas

We previously reported that morpholino silencing of *adrb1* rescued CVP cavernomas in zebrafish embryos (*Li et al., 2021a*). Because of morpholinos potential for off target effects (*Robu et al., 2007*; *Eisen and Smith, 2008*), we sought to confirm the β1AR's potential involvement in CCM pathogenesis by inactivating *adrb1*. We used CRISPR-Cas9 to generate an 8 bp deletion resulting in a premature stop codon at 57 bp (*Figure 1A*). To exclude potential off-target effects, the top 20 potential off-target sites predicted by Cas-OFFinder (https://www.rgenome.net/cas-offinder) (*Supplementary file 2*) were sequenced and were not mutated (data not shown). We intercrossed *adrb1*$^{+/-}$ +/-1 offspring resulting in *adrb1*$^{-/-}$ zebrafish embryos that displayed reduced cardiac contractility (*Videos 1 and 2*) and decreased heart rate (*Figure 1—figure supplement 1A*). Similar to *Adrb1*$^{-/-}$ mice (*Rohrer et al., 1996*), *adrb1*$^{-/-}$ zebrafish exhibited a blunted chronotropic response to a β1AR agonist, isoprenaline hydrochloride (*Figure 1B*). *Adrb1*$^{-/-}$ embryos had no obvious defects in vascular development (*Figure 1—figure supplement 2*) and they survived to adulthood and were fertile.

We intercrossed *adrb*$^{+/-}$ +/-and injected one-cell stage embryos with *ccm2* CRISPR and blindly scored the presence of CVP cavernomas at 48hpf. As expected we observed CVP cavernomas in 28% of *adrb1*$^{+/+}$ embryos. In sharp contrast only 3% of *adrb1*$^{-/-}$ embryos exhibited cavernomas (*Figure 1C and D*) indicating that loss of β1AR prevents CVP cavernomas. these observations demonstrate that the β1AR is required for embryonic CVP cavernoma formation.

### β1AR mediates formation of adult CCM in the brain

Adult *ccm2* CRISPR zebrafish display highly penetrant CCMs throughout the central nervous system (*Li et al., 2021b*). Brains from adult *ccm2* CRISPR fish on *adrb1*$^{-/-}$ (12 brains) or wild type (13 brains) background were treated with CUBIC (clear, unobstructed brain/body imaging cocktails and computational analysis; *Susaki et al., 2015*), and these transparent brains were then scanned with light-sheet microscopy and lesions were enumerated and volumes were estimated with NIH ImageJ. While the typical multi-cavern CCMs were observed in brains on wild type background appearing as blood filled dilated vessels (*Figure 2A* through C), *ccm2* CRISPR *adrb1*$^{-/-}$ fish exhibited an 87% reduction in lesion volume (*Figure 2D* through G). Thus, genetic inactivation of β1AR prevented CCMs.

### A selective β1AR antagonist prevents CCMs

The non-selective β blocker propranolol reduces lesion volume in murine CCM models (*Li et al., 2021a*; *Oldenburg et al., 2021*). To ascertain whether propranolol had a similar effect in the zebrafish model, the chemical treatment was started from larval stage at concentration which allows the fish

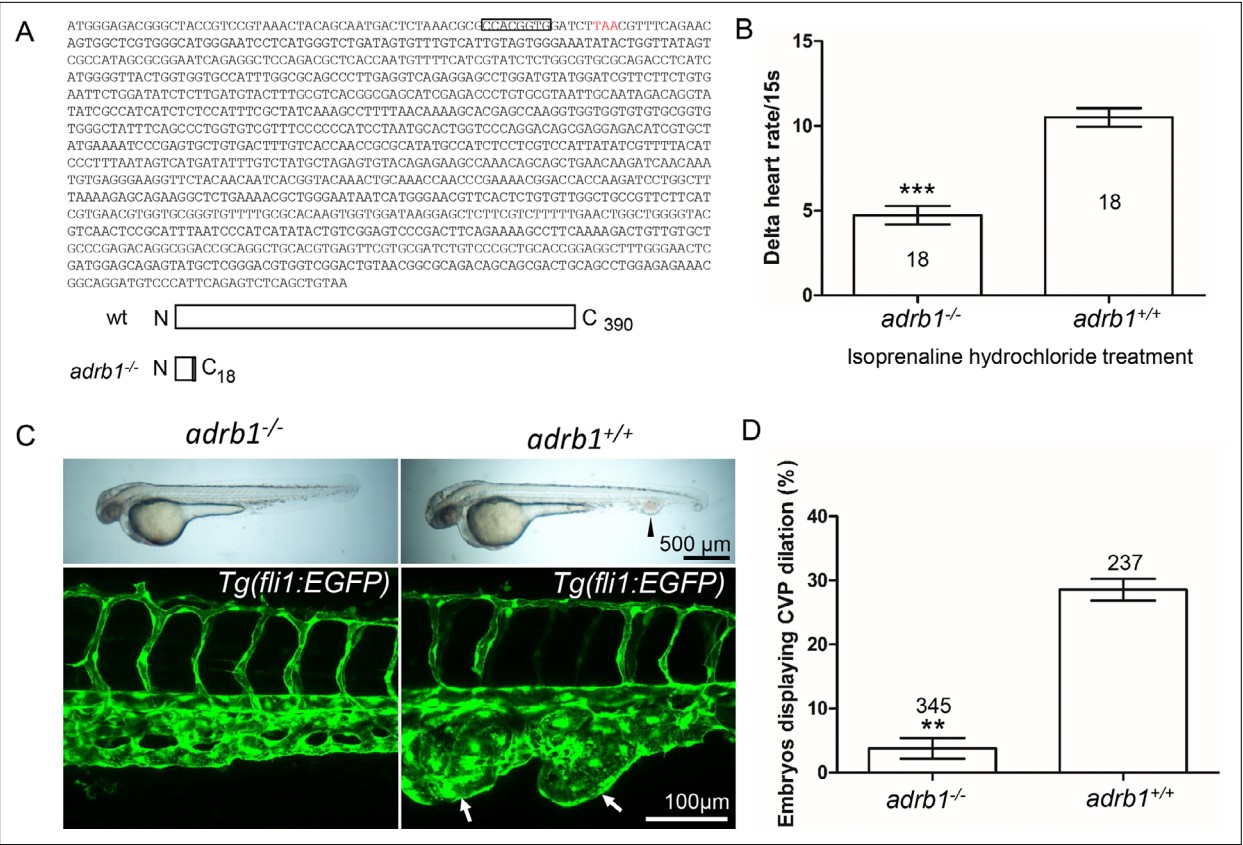

**Figure 1.** Adrb1 signaling is essential for CVP dilation. (**A**) The targeted *adrb1* allele shows an 8-nucleotide deletion producing a pre-stop codon. *Adrb1* null cDNA is predicted to encode truncated adrb1 protein. The wild type adrb1 protein contains 390 amino acids, while the predicted adrb1 null protein would contain 2 missense amino acids (gray bar) and would terminate after amino acid 18. (**B**) Isoprenaline hydrochloride (50μM) treatment at 72hpf lead to a heart rate increase in zebrafish, while the delta heart rate in *adrb1*−/− is significantly smaller than that of wild type. Heartbeat was counted in 18 embryos of each group before and immediately after adding the chemical. Paired two-tailed t test, p<0.0001. (**C**) After *ccm2* CRISPR injection, representative bright field and confocal images of 2dpf *Tg(fli1:EGFP)* embryos show that wild type embryos display CVP dilation, while *adrb1*−/− embryos were resistant to this defect. Arrowhead and arrows indicate the dilation in CVP. Scale bar: 500 μm (bright field), 100 μm (confocal). (**D**) Paired two-tailed t test shows that percentage of embryos displaying CVP dilation is significantly smaller on *adrb1*−/− background than that of control embryos. p=0.0012. 345 *adrb1*−/− embryos and 237 control embryos from four experiments were examined for CVP cavernoma.

The online version of this article includes the following figure supplement(s) for figure 1:

**Figure supplement 1.** *Adrb1*−/− zebrafish embryos displayed a decrease of heart rate and blood flow in CVP.

**Figure supplement 2.** No significant difference of CVP development was observed between *adrb1*−/− and wild type embryos.

to develop to two months for CCM inspection. We added 12.5 μM propranolol to or vehicle control to fish water of *ccm2* CRISPR zebrafish starting at 3 weeks of age. The water was refreshed daily with drug or vehicle until fish were sacrificed and brains were examined as described (*Figure 3A*). Quantification based on light-sheet scanning of zebrafish brains (*Figure 3B*) showed that compared to vehicle-treated controls (*Figure 3C, D and E*), propranolol-treated groups displayed a 94% reduction in CCM lesion volume (*Figure 3F, G and H*). Similarly, administration of 50 μM racemic metoprolol a β1-selctive antagonist produced a similar (98%) reduction in lesion volume (*Figure 3B, I, J and K*). Importantly, neither drug at the doses administered reduced the growth of the fish or the volume of their brains. In sum, both genetic and pharmacological loss of β1 adrenergic receptor signaling markedly reduces the lesion burden in the zebrafish *ccm2* CRISPR model of CCM.

## Loss of β1AR does not prevent increased *klf2a* expression in *ccm2* null embryos

Inactivation of CCM genes leads to increased endothelial KLF2 expression (*Renz et al., 2015*; *Zhou et al., 2015*), a transcription factor important in cavernoma formation (*Zhou et al., 2016*; *Li et al.,*

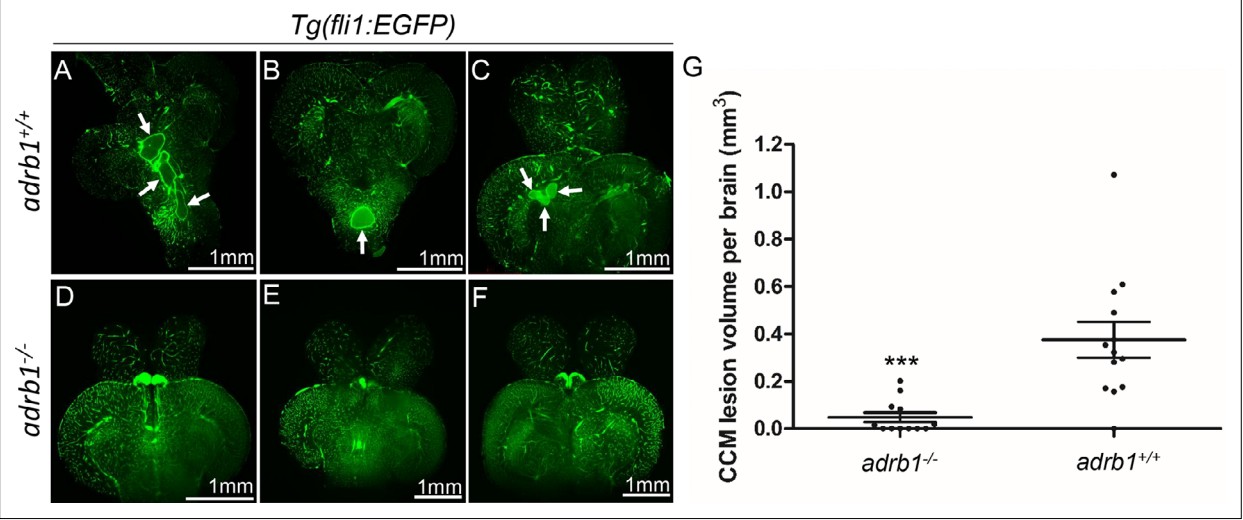

**Figure 2.** Genetic inhibition of adrb1 signaling could rescue CCM in *ccm2* CRISPR zebrafish. (**A** through **F**) Representative light sheet microscopy scanning pictures of brains from *ccm2* CRISPR adult zebrafish of *adrb1*$^{+/+}$ (**A** through **C**) and of *adrb1*$^{-/-}$ (**D** through **F**) on *Tg(fli1:EGFP)* background. Brains from *ccm2* CRISPR on wild type background show lesions indicated by arrows (**A** through **C**), while brains from *ccm2* CRISPR on *adrb1*$^{-/-}$ do not show lesions (**D** through **F**). Scale bar: 1 mm. (**G**) Statistical analysis of total lesion volume by unpaired two-tailed t test. p=0.0005. 12 *adrb1*$^{-/-}$ brains and 13 *adrb1*$^{+/+}$ brains were analyzed.

*2021b*). Nevertheless, silencing of *adrb1* did not prevent the expected increased endothelial *klf2a* (*Renz et al., 2015*) expression in *ccm2* morphant *Tg(klf2a:H2b-EGFP)* fish in which the nuclear EGFP expression is driven by the *klf2a* promoter (*Figure 4A, B and C*). We previously reported that mosaic expression of KLF2a occurs in *tnnt* morphant 2 dpf *ccm2* CRISPR embryos, as judged by a widely variable in endothelial *klf2a* reporter expression (*Li et al., 2021b*). Nevertheless, combination of *tnnt2a* morphant with the *adrb1* morphant (*Figure 4D*) or control morphant (*Figure 4E*) *ccm2* CRISPR *Tg(klf2a:H2b-EGFP)* embryos both displayed a similar widely variable *klf2a* reporter intensity (*Figure 4F*) indicative of similar mosaicism. Similarly, merely slowing the heart and reducing contractility with 2,3-butanedione monoxime (BDM; *Bartman et al., 2004*) also prevented CVP cavernomas (*Figure 4G*, *Figure 4—figure supplement 1*). Thus, silencing β1AR does not prevent the generalized increase in endothelial KLF2a in *ccm2* morphants nor does it prevent the mosaic KLF2a increase in *ccm2* CRISPR zebrafish.

## Discussion

In this study, we provide genetic and pharmacological evidence to implicate the β1 adrenergic receptor (β1AR) in the pathogenesis of CCM. Our data suggest that inactivating β1AR via gene inactivation or pharmacological inhibition can significantly reduce the volume of CCM lesions in a zebrafish model.

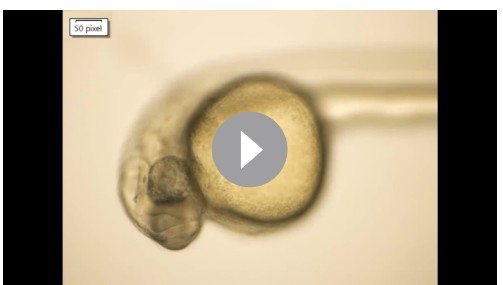

**Video 1.** The cardiac pumping in *adrb1*$^{-/-}$ embryos at 28hpf.
https://elifesciences.org/articles/99455/figures#video1

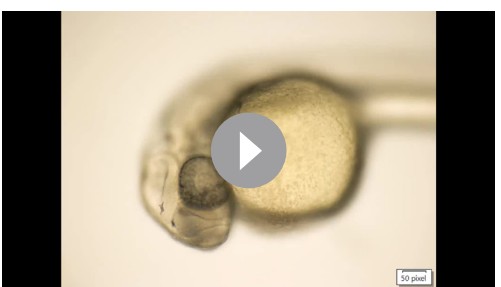

**Video 2.** The cardiac pumping in wild type embryos at 28hpf.
https://elifesciences.org/articles/99455/figures#video2

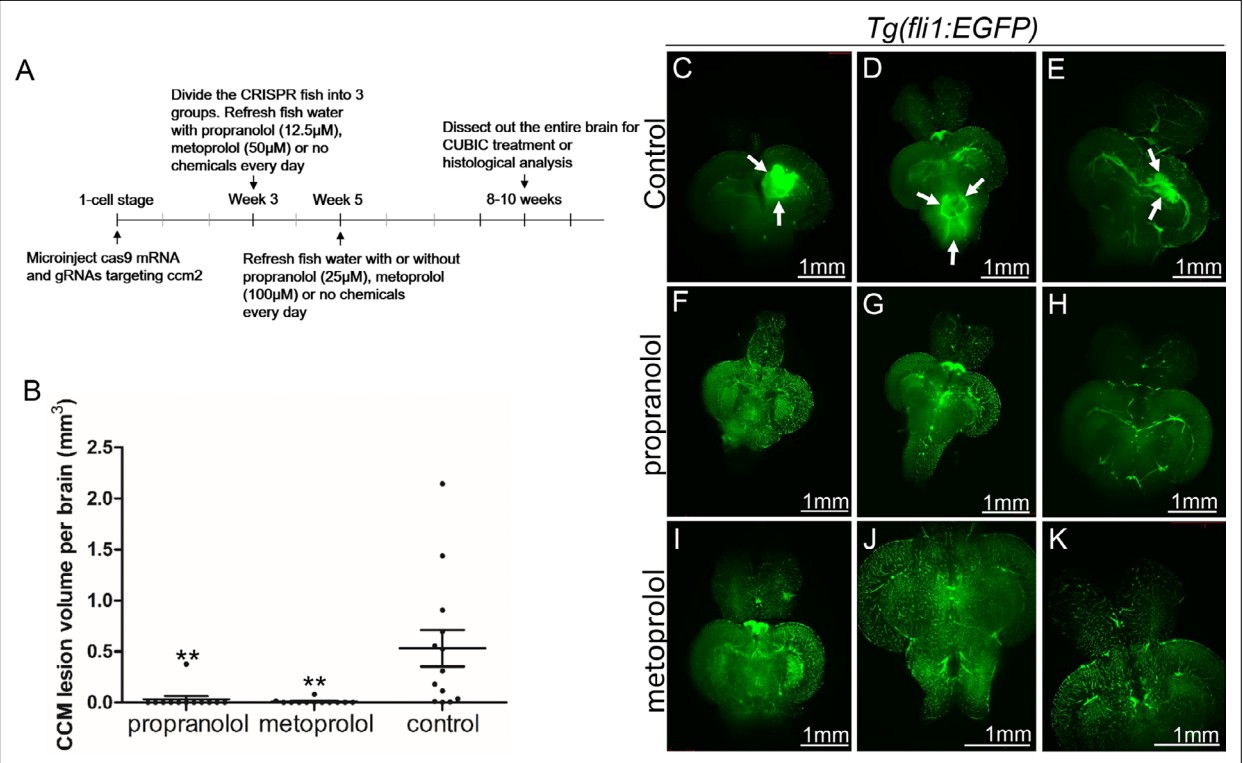

**Figure 3.** Both propranolol and metoprolol could rescue CCM in *ccm2* CRISPR zebrafish. (**A**) A diagram outlines the drug treatment experiment, CUBIC treatment and following recording of CCMs in adult zebrafish brain. The chemical treatment was started from week 3 with 12.5 µM propranolol or 50 µM metoprolol, and increased to 25 µM propranolol and 100 µM metoprolol, respectively from week 5. The fish water with chemicals or vehicle control are refreshed on a daily basis. (**B**) Statistical analysis of lesion volume by one-way ANOVA followed by Tukey's multiple comparison test. p<0.01. 12 propranolol treated, 12 metoprolol treated, and 13 vehicle brains were analyzed. (**C** through **K**) Representative light sheet microscopy scanning pictures of brains from *ccm2* CRISPR adult zebrafish on *Tg(fli1:EGFP)* background. In controls without chemical treatment (**C**, **D**, and **E**) there were vascular anomalies indicated by arrows. Neither propranolol (**F**, **G**, and **H**) nor metoprolol (**I**, **J**, and **K**) treated fish showed vascular lesions in the brain. Scale bar: 1 mm.

Case reports (*Moschovi et al., 2010*; *Miquel et al., 2014*; *Reinhard et al., 2016*; *Goldberg et al., 2019*) and our previous study (*Li et al., 2021a*) revealed the potential benefit of the non-selective β-adrenergic receptor blocker propranolol in reducing CCMs in patients and mouse models, respectively. However, propranolol is a racemic mixture, and its R enantiomer which lacks β-adrenergic antagonism was reported to show therapeutic effect for infantile hemangiomas in an animal study (*Overman et al., 2019*). Notably, our study demonstrates that *adrb1*−/− zebrafish are significantly protected against the formation of CCMs, suggesting that propranolol's therapeutic effect on CCM is through β1AR antagonism. Together with the observed significant reduction of CCM lesion volume upon metoprolol treatment, a selective β1AR antagonist, supports the therapeutic potential of β1AR antagonism in CCMs. β1 selective blockers offer the advantage of causing fewer mechanism-based side effects compared to propranolol, such as bronchospasm (*Ji et al., 2018*), and are already in clinical use (including agents like atenolol, metoprolol, nebivolol, and bisoprolol). Thus, further studies are warranted to evaluate the potential value of β1 selective antagonists in this disease.

Consistent with the *Adrb1KO* mice (*Rohrer et al., 1996*), the *adrb1*−/− zebrafish embryos displayed the decreased chronotropic response to a beta-adrenergic agonist, isoprenaline, and decreased basal heart rate compared to that of *adrb1*+/+ brood mates; however, CVP morphology of *adrb1*−/− embryo was not perturbed (*Figure 1—figure supplement 2*). *adrb1*−/− embryo also displayed a marked decrease of blood flow in CVP (*Videos 3 and 4*, and *Figure 1—figure supplement 1B*), A similar reduction of heart rate and contractility by 2,3-BDM, a cardiac myosin ATPase inhibitor, prevented CVP dilation (*Figure 4F*). We previously found that arresting blood flow prevents aberrant intussusceptive angiogenesis and the resulted CVP cavernomas in *ccm2* CRISPR embryos (*Li et al., 2021b*). Taken together, these data suggest that reduced blood flow secondary to reduced cardiac function accounts for the

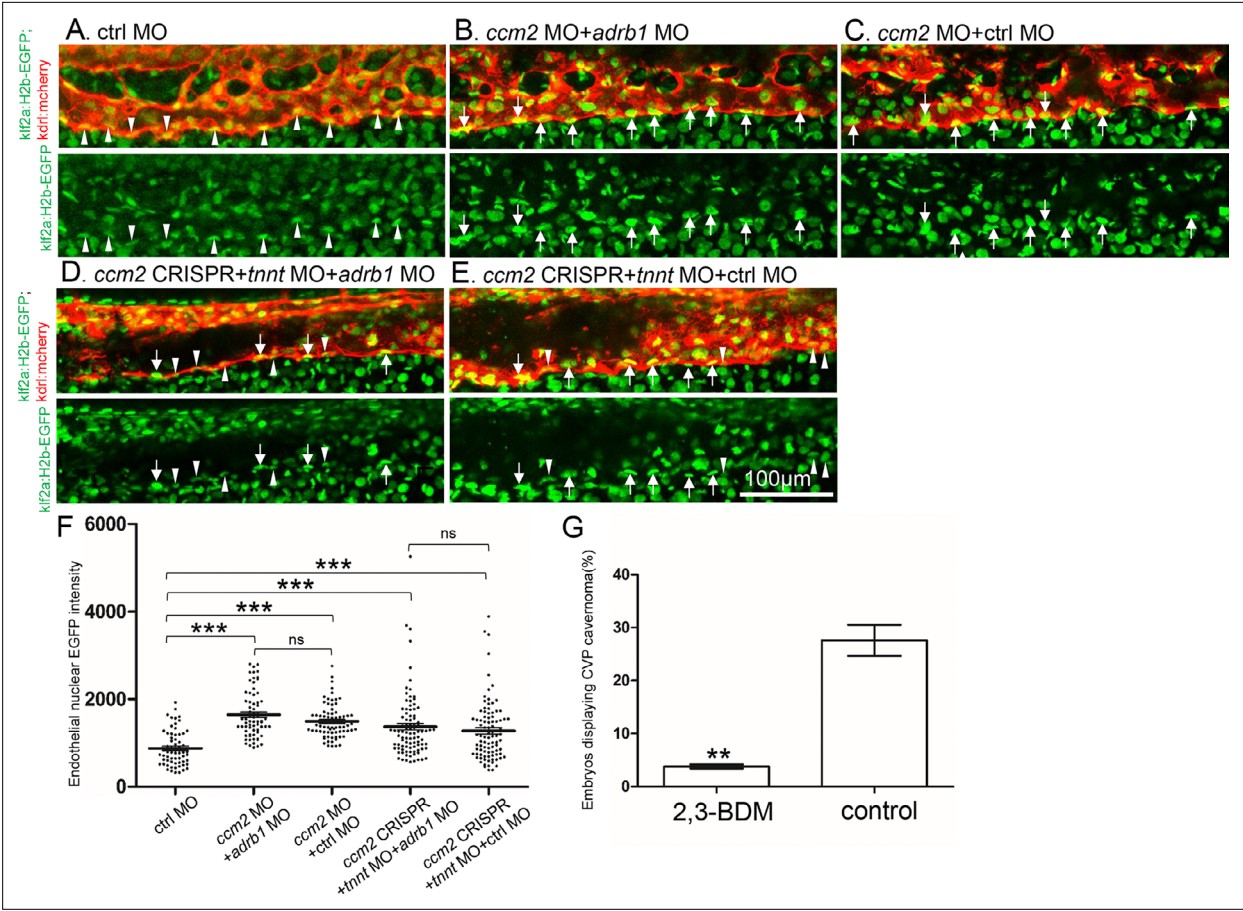

**Figure 4.** Adrb1 signaling does not alter *klf2a* expression in *ccm2* CRISPR embryos. *Tg(klf2a:H2b-EGFP; kdrl:mcherry)* embryos were injected and nuclear EGFP signal in mcherry labeled vascular endothelial cells is recorded by confocal. Representative images from each group are shown. (**A**) Control MO alone injected embryos were used as control. (**B and C**) *Ccm2* morphant embryos co-injected with *adrb1* MO (**B**) or control MO (**C**) both displayed significant increase of endothelial nuclear EGFP intensity (p<0.0001) compared to that of control (**A**), and there is no significant difference between them. (**D and E**) All the *ccm2* CRISPR embryos were co-injected with *tnnt* MO, which are absent of blood flow. Compared to that of control (**A**), *ccm2* CRISPR embryos co-injected with *adrb1* MO (**D**) or control MO (**E**) both displayed a mosaic increase of nuclear EGFP intensity of vascular endothelial cells compared to control (**A**) (<0.0001), and there is no significant difference between them. Arrows indicated the endothelial nuclei with significant higher EGFP intensity than those indicated by arrowheads. Scale bar:100 μm. (**F**) EGFP intensity of endothelial nuclei were quantified with Image J. The number of analyzed nuclei were: 63 from 10 embryos (control MO), 70 from 10 embryos (*ccm2* MO + *adrb1* MO), 77 from 10 embryos (*ccm2* MO + control MO), 93 from 13 embryos (*ccm2* CRISPR +*adrb1* MO), and 94 from 13 embryos (*ccm2* CRISPR +control MO). Statistical analysis is performed by one-way ANOVA followed by Tukey's multiple comparison test. (**G**) At 2dpf, 2,3-BDM prevented the CVP cavernoma dramatically. 164 embryos in 2,3-BDM treated group and 177 in control group were used for Two-tailed paired t-test. p=0.0013.

The online version of this article includes the following figure supplement(s) for figure 4:

**Figure supplement 1.** 2,3-BDM decreases the heart rate in 30hpf zebrafish embryos.

protective effect of loss of β1AR signaling on CVP lesions and on CCM. To further confirm that reduced blood flow underlies the role of β1AR antagonism in rescuing these vascular defects, chemicals such as cardiac glycosides or phosphodiesterase inhibitors could be employed to restore cardiac pumping function in *adrb1⁻/⁻* embryos. Importantly, although β1AR are highly expressed in cardiomyocytes and contribute to increased cardiac output (*Rohrer et al., 1999*), β1ARs are also expressed in other tissues (*Osswald and Guimarães, 1983*; *Guimarães and Moura, 2001*) including endothelial cells of vascular anomalies in patients (*Stănciulescu et al., 2021*). Thus, future studies will be required to delineate the tissue-specific contributions of β1AR signaling to CCM pathogenesis.

# Materials and methods

## Zebrafish lines and handling

Zebrafish were maintained and with approval of Institutional Animal Care and Use Committee of the University of California, San Diego. The following mutant and transgenic lines were maintained under standard conditions: *Tg(fli1:EGFP)[y1]* (*Lawson and Weinstein, 2002*), *Tg(klf2a:H2b-EGFP)* (*Heckel et al., 2015*), and *Tg(kdrl:mcherry)[is5]* (*Wang et al., 2010*). *adrb1[-/-]* zebrafish was obtained by co-injection of Cas9 protein (EnGen Spy Cas9 NLS, M0646, NEB) with gRNA targeting *adrb1*. Genotyping of *adrb1[-/-]* was performed with forward primer (5'-AGAGCAGAGCGCGGATGGAA-3') and reverse primer (5'-GATCCATACATCCAGGCT-3').

## Plasmids and morpholino

pCS2-nls-zCas9-nls (47929) and pT7-gRNA (46759) were bought from Addgene. The CRISPR RNA (crRNA) sequences used in this study are as follow: *ccm2*-1 5'-GGTGTTTCTGAAAGGGGAGA-3', *ccm2*-2 5'- GGAGAAGGGTAGGGATAAGA-3', *ccm2*-3 5'-GGGTAGGGATAAGAAGGCTC-3', *ccm2*-4 5'-GGACAGCTGACCTCAGTTCC-3', adrb1 5'-GACTCTAAACGCGCCACGG-3'. Target gRNA constructs were generated as described before (*Jao et al., 2013*). Morpholino sequence used in this study are: *adrb1* (5'-ACGGTAGCCCGTCTCCCATGATTTG-3') (*Steele et al., 2011*), *ccm2* (5'-GAAG CTGAGTAATACCTTAA CTTCC-3') (*Mably et al., 2006*), *tnnt2a* (5'-CATGTTTGCTCTGATCTGAC ACGCA-3')(*Sehnert et al., 2002*), control (5'- CCTCTTACCTCAGTTACAATTTATA-3').

## RNA synthesis

The pCS2-nls-zCas9-nls plasmid containing Cas9 mRNA was digested with NotI enzyme, followed by purification using a Macherey-Nagel column, serving as the template. The capped nls-zCas9-nls RNA was synthesized using the mMESSAGE mMACHINE SP6 Transcription Kit from ThermoFisher Scientific. The resulting RNA was purified through lithium chloride precipitation, as per the instructions provided in the kit. For gRNA synthesis, the gRNA constructs were linearized using BamHI enzyme and purified using a Macherey-Nagel column. The gRNA was synthesized via in vitro transcription using the MEGAshortscript T7 Transcription Kit from ThermoFisher Scientific. After synthesis, the gRNA was purified by alcohol precipitation, as instructed in the same kit. The concentration of the nls-zCas9-nls RNA and gRNA was measured using a NanoDrop 1000 Spectrophotometer from Thermo Fisher Scientific, and their quality was confirmed through electrophoresis on a 1% (wt/vol) agarose gel.

## Microinjection

All injections were performed at 1-cell stage with a volume of 0.5 nl. The final injection concentrations are as follow: Cas9 protein (10 µM), Cas9 mRNA (750 ng/µl), gRNA 120 ng/µl, *adrb1* MO (4 ng/µl), *ccm2* MO (4 ng/µl), *tnnt2a* MO (5.3 ng/µl), control MO (4 ng/µl).

## Chemical treatment

Propranolol (P0995, TCI; 12.5 µM) and metoprolol (M1174, Spectrum; 50 µM) were used to treat the zebrafish larva from Day 21. Beginning from Day 35, adjusted concentration of propranolol (25 µM) or metoprolol (100 µM) were used to treat the juvenile fish. The chemicals were dissolved in fish water,

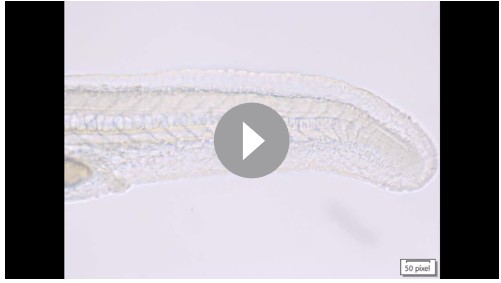

**Video 3.** The blood flow in CVP in *adrb1[-/-]* embryos at 28hpf.

https://elifesciences.org/articles/99455/figures#video3

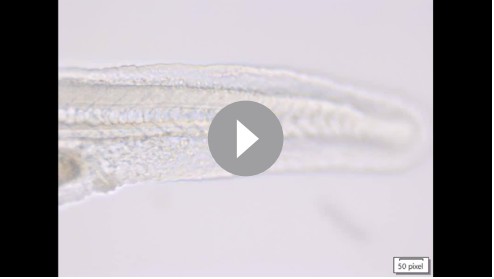

**Video 4.** The blood flow in CVP in wild type embryos at 28hpf.

https://elifesciences.org/articles/99455/figures#video4

and fish water containing chemicals were refreshed daily. Egg water without above chemicals was refreshed daily for fish used as negative control. 2,3- butanedione monoxime (BDM; 6 mM) was added to egg water of the *ccm2* CRISPR embryos at 25hpf, and CVP cavernoma was observed at 2dpf.

### Airyscan imaging and fluorescence intensity analysis

To prepare the embryos for imaging, they were first anesthetized using egg water containing 0.016% tricaine (3-amino benzoic acid ethyl ester) from Sigma-Aldrich. Subsequently, the anesthetized embryos were embedded in 1% low melting point agarose obtained from Invitrogen (product number 16520050). The imaging process was carried out using a Zeiss 880 Airyscan confocal microscope, utilizing the standard Airyscan stack mode, with a Plan-Apochromat 20 x/0.8 M27 objective. The scanning setup is as follow: Lasers (Green 488 nm: 26.0%, Red 561 nm: 15.0%), Master Gain (800), Digital Gain (1.00), Scaling X (0.415 μm), Scaling Y (0.415 μm), and Scaling Z (0.800 μm). The intensity of nuclear EGFP (enhanced green fluorescent protein) was quantified using ImageJ software. The selected background area signal was measured by running Analyze >Measure. Then the background value was subtracted by running Process >Math > Subtract. "Freehand selections" button was used for outlining the endothelial nucleus stack by stack along Z-axis. By running Analyze >Measure, the information of "Area" and "IntDen (Integrated Density)" of the selected endothelial nucleus was obtained. The average EGFP intensity of a nucleus equals to the summation of "IntDen" divided by summation of "Area".

### Heartbeat and blood flow recording

Heartbeat and blood flow were recorded using Olympus MVX10. The embryos were treated with 0.004% tricaine which does not have effect on heartbeat (*Langheinrich et al., 2003*; *Schwerte et al., 2003*).

### Zebrafish brain dissection, CUBIC treatment and lightsheet imaging

The dissection of zebrafish brains followed the methodology described in a previous study by *Gupta and Mullins, 2010*. The CUBIC method was optimized based on the findings from a previous report (*Susaki et al., 2015*). The brains were fixed in 4% paraformaldehyde (PFA) with a pH of 7.5 for 24 hr and subsequently washed with PBS (phosphate-buffered saline) for an additional 24 hr. Following the PBS wash, the brains underwent CUBICR1 treatment at 37 °C in a water bath for 42 hr. For imaging, the samples were placed in CUBICR2 as the imaging medium and imaged using a ZEISS Lightsheet Z.1 microscope. Scanning was carried out utilizing 5 X dual illumination optics in combination with a 5 X objective.

### Statistical analysis

The statistical analysis was conducted using GraphPad Prism software. p-Values were calculated using an unpaired two-tailed Student's t-test, unless otherwise specified. The bar graphs display the mean values along with their corresponding SEM (standard error of the mean) error bars.

## Acknowledgements

We gratefully acknowledge Douglas A Marchuk, and Issam A Awad for their invaluable advice on both this study and the manuscript. This work was supported by NIH grants P01 NS092521 and P01 HL151433. We also acknowledge Jennifer Santini and Marcy Erb for microscopy technical assistance and resources provided by the UCSD School of Medicine Microscopy Core (NINDS P30 NS047101).

## Additional information

### Funding

| Funder | Grant reference number | Author |
| --- | --- | --- |
| NIH | P01 NS092521 | Mark H Ginsberg |

| Funder | Grant reference number | Author |
|--------|------------------------|--------|
| NIH | P01 HL151433 | Mark H Ginsberg |

The funders had no role in study design, data collection and interpretation, or the decision to submit the work for publication.

## Author contributions

Wenqing Li, Conceptualization, Data curation, Software, Formal analysis, Supervision, Validation, Investigation, Visualization, Methodology, Writing – original draft, Project administration, Writing – review and editing; Sara McCurdy, Miguel A Lopez-Ramirez, Software; Ho-Sup Lee, Data curation; Mark H Ginsberg, Resources, Funding acquisition, Writing – review and editing

## Author ORCIDs

Wenqing Li ⓘ https://orcid.org/0000-0002-2721-8603

## Ethics

All animal procedures were approved by the Institutional Animal Care and Use Committee (IACUC) of the University of California, San Diego (Protocol S14135). All efforts were made to minimize animal suffering and ensure ethical standards in accordance with institutional and federal guidelines.

Reviewer #2 (Public review): https://doi.org/10.7554/eLife.99455.3.sa1
Author response https://doi.org/10.7554/eLife.99455.3.sa2

# Additional files

## Supplementary files

Supplementary file 1. Comparison of the two-phase zebrafish CCM model with mouse CCM model and human CCM. "?" means it is yet to be determined. "-" means it is not applicable. "1". Perilesional red blood cell leakage was seen.

Supplementary file 2. The predicted off-targets genomic sites produced by adrb1 CRISPR. These genomic sites were sequenced and found no mutations. Primer sequence used for amplifying these sites were listed.

MDAR checklist

## Data availability

Raw images for the figures of this manuscript have been deposited with Dryad (https://doi.org/10.5061/dryad.dz08kps7n). Raw phenotype counts have been provided in figures and figure legends.

The following dataset was generated:

| Author(s) | Year | Dataset title | Dataset URL | Database and Identifier |
|-----------|------|---------------|-------------|-------------------------|
| Li W | 2025 | Raw images from: Genetic inactivation of the β1 adrenergic receptor prevents Cerebral Cavernous Malformations in zebrafish | https://doi.org/10.5061/dryad.dz08kps7n | Dryad Digital Repository, 10.5061/dryad.dz08kps7n |

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
