## [Editor Report · eLife Assessment]

In this **important** study, the authors test the model that a type of vascular lesion caused by the inactivation of one gene in the cells that line blood vessels requires the activity of a second gene for the lesions to form. The evidence supporting the conclusions is **solid**.

---

## [Referee Report · Reviewer #2 (Public review)]

Summary:

Previously, the authors developed a zebrafish model for cerebral cavernous malformations (CCMs) via CRISPR/Cas9-based mosaic inactivation of the ccm2 gene. This model yields CCM-like lesions in the caudal venous plexus of 2 days post-fertilization embryos and classical CNS cavernomas in 8-week fish that depend, like the mouse model, on the upregulation of the KLF2 transcription factor. Remarkably, the morpholino-based knockdown of the gene encoding the Beta1 adrenergic receptor or B1AR (adrb1; a hemodynamic regulator) in fish and treatment with the anti-adrenergic S enantiomer of propranolol in both fish and mice reduce the frequency and size of CMM lesions.

In the present study, the authors aim to test the model that adrb1 is required for CCM lesion development using adrb1 mutant fish rather than morpholino-mediated knockdown and pharmacological treatments with the anti-adrenergic S enantiomer of propranolol or a racemic mix of metoprolol (a selective B1AR antagonist).

Strengths:

The goal of the work is important, and the findings are potentially highly relevant to cardiovascular medicine.

Comments on latest version:

This reviewer is largely satisfied and congratulates the authors on their updated work. However, the comments regarding the caveats of morpholino use and lack of validation that the morphants phenocopy the mutants using the readouts that they employ still stand (for instance, the tnnt2a MO has been extensively validated for phenocopying lack of cardiac contractility, not for the phenotypes under study). Finally, while using the cytosolic red line to mask a nuclear green readout is suboptimal (not for FRET reasons), this is now a minor issue given that all comparisons are made using this method and the increase in sample size.

---

## [Author Response]

The following is the authors’ response to the original reviews.

**Reviewer #1 (Public review):**
Summary:This work seeks to provide genetic evidence for a role for beta-adrenergic receptors that regulate heart rate and blood flow on cavernous malformation development using a zebrafish model, and to extend information regarding beta-adrenergic drug blockade in cavernous malformation development, with the idea that these drugs may be useful therapeutically.Strengths:The work shows that genetic loss of a specific beta-adrenergic receptor in zebrafish, adrb1, prevents embryonic venous malformations and CCM in adult zebrafish brains. Two drugs, propranolol and metoprolol, also blunt CCM in the adult fish brain. These findings are predicted to potentially impact the treatment of human CCM, and they increase understanding of the factors leading to CCM.

Response 1: We are grateful for the reviewer’s acknowledgment of this study’s potential translational significance.

Weaknesses:There are minor weaknesses that detract slightly from enthusiasm, including poor annotation of the Figure panels and lack of a baseline control for the study of Klf2 expression (Figure 4).

Response 2: We agree. Annotation of the Figure panels were added, and a baseline control for the study of *klf2a* expression (Figure 4) was added. Details were described in the response to “recommendations for the authors”.

**Reviewer #2 (Public review):**
Summary:Previously, the authors developed a zebrafish model for cerebral cavernous malformations (CCMs) via CRISPR/Cas9-based mosaic inactivation of the ccm2 gene. This model yields CCM-like lesions in the caudal venous plexus of 2 days post-fertilization embryos and classical CNS cavernomas in 8-week fish that depend, like the mouse model, on the upregulation of the KLF2 transcription factor. Remarkably, the morpholino-based knockdown of the gene encoding the Beta1 adrenergic receptor or B1AR (adrb1; a hemodynamic regulator) in fish and treatment with the anti-adrenergic S enantiomer of propranolol in both fish and mice reduce the frequency and size of CMM lesions.In the present study, the authors aim to test the model that adrb1 is required for CCM lesion development using adrb1 mutant fish rather than morpholino-mediated knockdown and pharmacological treatments with the anti-adrenergic S enantiomer of propranolol or a racemic mix of metoprolol (a selective B1AR antagonist).Strengths:The goal of the work is important, and the findings are potentially highly relevant to cardiovascular medicine.

Response 3: We are grateful for the reviewer’s acknowledgment of this study’s scientific importance and clinical relevance.

Weaknesses:(1) The following figures do not report sample sizes, making it difficult to assess the validity of the findings: Figures 1B and D (the number of scored embryos is missing), Figures 2G and 3B (should report both the number of fish and lesions scored, with color-coding to label the lesions corresponding to individual fish in which they were found).

Response 4: We agree. Sample sizes of Figures 1B and D were added in the figures and figure legends. Sample sizes of Figures 2G and 3B were added in their figure legend respectively. The lesion volume in Figures 2G and 3B is the total lesion volume in each brain.

(2) Figure 4 has a few caveats. First, the use of adrb1 morphants (rather than morphants) is at odds with the authors' goal of using genetic validation to test the involvement of adrb1 in CCM2-induced lesion development.

Second, the authors should clarify if they have validated that the tnnt (tnnt2a) morpholino phenocopies tnnt2a mutants in the context in which they are using it (this reviewer found that the tnnt2a morpholino blocks the heartbeat just like the mutant, but induces additional phenotypes not observed in the mutants).

Response 5: We appreciate the reviewer’s comments; however, generating *adrb1-/-* and *tnnt2a-/- klf2a* reporter fish, while also ensuring the presence of only one EGFP transgene allele for intensity measurement, would require prohibitively time-consuming breeding efforts.

The use of morpholinos for *tnnt2a* and *adrb1*, as well as their effects on the heart, have been well-documented in previous studies (Sehnert AJ et al., *Nat Genet.* 2002;31:106-10; Steele SL et al., *J Exp Biol.* 2011;214:1445-57).

Third, the data in Figure 4E is from just two embryos per treatment, a tiny sample size. Furthermore, judging from the number of points in the graph, only a few endothelial PCV cells appear to have been sampled per embryo. Also, judging from the photos and white arrowheads and arrows (Figure 4A-D), only the cells at the ventral side of the vessel were scored (if so, the rationale behind this choice requires clarification).

Response 6: We have increased the sample size, as described in the Figure 4 legend. Regarding the scoring of endothelial nuclei, we focused on the ventral side of the vessel because nuclei on the dorsal side often reside at branching points of the venous plexus. This positional variance could influence *klf2a* expression levels; thus, we focused on the ventral surface to limit this potential confounding variable.

Fourth, it is unclear whether and how the Tg(kdrl:mcherry)is5 endothelial reporter was used to mask the signals from the klf2a reporter. The reviewer knows by experience that accuracy suffers if a cytosolic or cell membrane signal is used to mask a nuclear green signal.

Response 7: We agree that it is theoretically possible for Förster resonance energy transfer (FRET) to occur, as the emission spectrum of EGFP (495-550 nm in our filter setup) overlaps with the absorption spectrum of mCherry. However, several factors reduce the likelihood of FRET in our experimental setup:

(1) Without a nuclear localization signal, the majority of mCherry is localized in the cytoplasm, although small amounts may passively diffuse into the nucleus.

(2) EGFP, on the other hand, is predominantly localized in the nucleus due to the presence of a nuclear localization signal.

(3) FRET requires two fluorophores to be within a proximity of 8-10 nanometers or less for efficient energy transfer. The nuclear envelope, with a typical thickness of 30-50 nanometers, separates nuclear EGFP from cytoplasmic mCherry and FRET efficiency is inversely proportional to the sixth power of the distance between donor and acceptor. Thus, the theoretical likelihood of significant energy transfer under these conditions is low.

To empirically examine potential FRET between nuclear EGFP and mcherry in our experiment setup, we scanned and scored the Tg(klf2a:H2b-EGFP; kdrl:mcherry) double transgenic embryos and Tg(klf2a:H2b-EGFP) embryos for EGFP intensity. The result is attached here:

**Author response image 1. sa2fig1:** 42 endothelial nuclei from 7 embryos were scored as described in the Experimental Procedures of the manuscript. Two tailed t test were performed. P=0.4529

Finally, the text and legend related to Figure 4 could be more explicit. What do the authors mean by a mosaic pattern of endothelial nuclear EGFP intensity, and how is that observation reflected in graph 4E? When I look at the graph, I understand that klf2a is decreased in C-D compared to A-B. Are some controls missing? Suppose the point is to show mosaicism of Klf2a levels upon ccm2 CRISPR. Don't you need embryos without ccm2 CRISPR to show that Klf2a levels in those backgrounds have average levels that vary within a defined range and that in the presence of ccm2 mosaicism, some cells have values significantly outside that range? Also, in 4A-D, what are the white arrowheads and arrows? The legend does not mention them.

Response 8: We have revised our description of Figure 4 to better convey that mosaic expression of KLF2a is evidenced by the wide variability of *klf2a* reporter intensity in endothelial cells in *ccm2* CRISPR embryos. A baseline control for the study of *klf2a* expression was added to Figure 4. The arrowheads and arrows in Figure 4A-D are explained in Figure 4 legends.

Given the practical relevance of the findings to cardiovascular medicine, increasing the strength of the evidence would greatly enhance the value of this work.
**Recommendations for the authors:**

**Reviewing Editor:**
Concerns about the labeling of figures and sample sizes should both be addressed, as detailed in the reviews, as this will be important to ensure the robustness of the claims.
**Reviewer #1 (Recommendations for the authors):**
Overall a strong research advance that provides rigorous genetic analysis and further drug testing in the zebrafish CCM model. There are some minor issues that, if addressed, would strengthen the work.Minor issues:(1) Figures in general are very poorly annotated and labeled. None of the images in Figures 1-3 show the reporter used to visualize vessels/CM, and the scale bars are not sized in the Figures or legends. Figure 1B is an experiment where the effects of a drug that increases heart rate are evaluated in mutants and controls, but the drug is not mentioned in the figure panel. Figure 1D shows the percentage of embryos with CVP dilation, but the graph and accompanying description does not define whether the percent is relative to the total embryos from the intercross or the percent of that category having the CVP dilation.

Response 9: Changes were made in Figures and Figure legends. The transgenic reporter line Tg(fli1:EGFP) was annotated in Figures 1-3. Scale bars were sized in the Figures and Figure legends. The chemical used for Figure 1B was annotated in the Figure. The percentage of CVP dilation in the graph was explained in the Figure legend.

(2) Figure 4 does not include baseline data in unmanipulated embryos scored at the same time to show the increase in Klf2 expression with mosaic ccm2 deletion. This is important as the result in E is interpreted as a lack of change in the increase.

Response 10: A baseline control for the study of *klf2a* expression in Figure 4 was added.

**Reviewer #2 (Recommendations for the authors):**
SUGGESTIONS FOR EXPERIMENTS, DATA, OR ANALYSES(1) For maximum rigor, in the Figure 4 experiment, use adrb1 mutants and tnnt2a (silent heart) mutants (or verify that the adrb1 and tnnt2a morpholinos faithfully copy the phenotype of interest). See: Guidelines for morpholino use in zebrafish (PMID: 29049395; PMCID: PMC5648102).

Response 11: See Response 5.

(2) Increase sample sizes if appropriate.

Response 12: In the revised version of the manuscript, we have increased the sample size, as described in the Figure 4 legend.

(3) The imaging and fluorescence intensity analysis methods require more detail for reproducibility's sake. Please provide this information. See as a guideline: Guillermo MarquésThomas PengoMark A Sanders (2020) Science Forum: Imaging methods are vastly underreported in biomedical research eLife 9:e55133.

Response 13: We added detailed procedures for the “Airyscan imaging and fluorescence intensity analysis” in the “Experimental Procedures”.

(4) I suggest further clarifying how inhibition of B1AR prevents cavernoma formation. Given that lesion formation is suppressed in adrb1 mutants (which have slow blood flow) and 2,3-BDM treatment (which also slows blood flow) has a similar effect, the beneficial effects of propranolol and metoprolol might be due to the slowing of blood flow via B1AR targeting rather than reflecting that B1AR is a critical component of the genetic circuit for cavernoma formation. Indeed, in prior work by the same first author and collaborators (Elife 2021 May 20:10:e62155), the investigators observed reduced cavernoma formation in embryos devoid of cardiac contractility and thus lacking blood flow (tnnt2a morphants). Such a scenario does not take away the value of a pharmacological treatment. Still, it implies a different mechanism and allows potentially many other drugs with similar effects on blood flow to be effective.Discussing how B1AR activity is regulated and outlining future experiments would be helpful. Suggestions for the latter include testing the effect of normalizing blood flow in adrb1 mutants with a drug or providing exogenous B1AR in the myocardium or the endothelium to test the model further.

Response 14: We are grateful for the reviewer’s suggestions and added the statement for future experiments.

MINOR CORRECTIONS TO TEXT AND FIGURES(1) Figure 4E: Label the four genotypes explicitly, rather than A-D for the reader's ease.(2) Legend of Figure 4: "(F) EGFP intensity...". It should be (E).CITATIONS TO CORRECT(1) The citation for the Tg(kdrl:mcherry)is5 transgene needs to be corrected (reference 29 is from the Stainier lab). However, the "is" designation is for the Essner lab (https://zfin.org/action/feature/view/ZDB-ALT-110127-25)

Response 15: Corrections were made as instructed.